# Linear Offset-Free Model Predictive Control in the Dynamic PLS Framework

**Ligang Hou [1], Ze Wu [2], Xin Jin [1],\* and Yue Wang [1]**

1   School of Information and Control Engineering, Liaoning Shihua University, Fushun 113001, China; houligang@126.com (L.H.); wangyue@lnpu.edu.cn (Y.W.)

2   Dushanzi Oil and Gas Transportation Branch Company, Petrochina West Pipeline Company, Karamay 833699, China; wuze@petrochina.com.cn

\*   Correspondence: jinxin@lnpu.edu.cn; Tel.: +86-135-004-31319

**Abstract:** This work addresses the model predictive control (MPC) of the offset-free tracking problem in the dynamic partial least square (DyPLS) framework. Firstly, state space MPC based on the DyPLS is proposed. Then, two methods are proposed to solve the offset-free problem. One is to reform the state space model as a velocity form. Another is to augment the state space model with a disturbance model and estimate the mismatch between system output and model output with an estimator. Both methods use the system output as a feedback in the control scheme. Hence, the offset-free tracking is guaranteed, and unmeasured step disturbance can be rejected. The results of two simulations demonstrate the effectiveness of proposed methods.

**Keywords:** partial least square; model predictive control; state space model; offset-free control

## 1. Introduction

The main concept of MPC is to use a model of the system to predict the future system output. MPC achieves its popularity in the process industry due to its ability to deal with multivariable systems and systems with hard and soft constraints. Early MPCs such as IDCOM and DMC are based on step or impulse models [1]. More general input-output models such as ARX, ARMAX and CARIMA models are used in generalized predictive control, as illustrated by Clarke et al. [2]. Muske and Rawlings [3] proposed an MPC implementation based on state space model. The state space approach provides a unified framework for discussion of the various predictive control algorithms and is well suited for stability analysis [4]. In addition, the input-output models can be realized as state space models [5]. Therefore, MPC based on state space models is useful as an implementation paradigm.

Although many cases have proved that MPC has many advantages, there is still a weakness that is necessary to discuss: when unmeasurable disturbances and plant-model mismatches exist, offset tracking performance cannot be achieved. A variety of successful algorithms have been proposed to solve this problem. Xue Wang et al. [6] improved the dynamic matrix control algorithm by applying disturbance model in the states of control system, and a Kalman filter was used to estimate unmeasurable disturbance, guaranteeing that the system realized offset-free control in the presence of unmeasurable disturbance. M. Askari et al. [7] developed a less computational method to achieve an offset-free MPC control system. In their method, an observer which is designed to estimate the disturbances and states is employed to eliminate the steady state error. As a result, the robustness of the closed loop system against step-like disturbances and noisy measurement is tremendously improved. Joel A. Paulson et al. [8] considered the linear system with two sources of additive bounded uncertainties on the states. One is for unknown, deterministic structural/parametric plant-model mismatch, the other is stochastic exogenous system disturbances. The proposed method used estimates

of the deterministic model uncertainties to modify the nominal state and input targets. It allowed for achieving offset-free tracking of the mean of the controlled variables. Betti, G. et al. [9] described the system state model in the so-called velocity form, where the state is composed by the state increments and the output error, while the manipulated variable is the control increment. The velocity form does not require the use of a state estimator and does not require the steady state target for the plant state and control variables to be computed.

Nowadays, the process of production is becoming larger and more complex, which means that the dimensions of the system are getting bigger. It is worth noting that one can obtain a large amount of operating data of these systems with the development of computation and storage techniques. Many of these data contain useful information about the system, and they are highly correlated. This promotes the development of a data-driven modeling method. Partial least square (PLS) has proved to be rewarding in the data-driven modeling field, and has been applied to many areas, such as quality prediction, process monitoring and chemometrics [10]. To handle the dynamic modeling problem, a variety of methods combining dynamic models with PLS have been proposed in recent decades. Yining Dong and S. Joe Qin [11] proposed a dynamic inner PLS model, in which an explicit dynamic inner model was given, and the inner model and outer model were made consistent at the same time. A method combining the autoregressive exogenous structure and PLS was proposed by Kaspar and Ray [12]. Qinghua Chi et al. [13] extended this method and discussed the relevant identification method in the inner PLS. Junghui Chen et al. [14] proposed another dynamic PLS framework with ARX model. In addition to these modeling method, many attempts have been made to put forward new control strategies that compromise the merits of PLS. Junghui Chen et al. [13] proposed a novel decoupling PID strategy with PLS. LÜ and Liang [15] proposed a multi-loop constrained MPC scheme. Jianhua Zhang [16,17] proposed robust control based on PLS. Tianyi Gao et al. [18] proposed a new intelligent MPC strategy in modified PLS framework, where iterative regression in model building and the large number of important undetermined parameters are avoided. Jin et al. [19] proposed an offset-free MPC in PLS framework which involves integral action in the controller and guaranteed offset-free tracking performance.

Tatjewski [20] summarized three main methods for offset-free MPC, one is with a state-space model and a measured state, another is with state observation and estimation, the third is with extended velocity form state-space model. The precondition of the first method is that state is measured. It is not suitable for DyPLS. While, the second and third method is suitable for dynamic method. In this paper, we will attempt to extend the state space MPC scheme to a DyPLS model. To get offset-free tracking performance, two methods are proposed. One is to reform the state space model as a velocity form. The other is to introduce an observer model in the control scheme. Both methods include output feedback in the control scheme. The rest of paper is organized as follows: State space MPC based on DyPLS and reasons for steady-state errors are described in Section 2. In Section 3, two offset-free state space MPCs based on Section 2 are proposed. In Section 4, two simulations are given to demonstrate the merit of the proposed method. In addition, conclusions are drawn in Section 5.

## 2. DyPLS Framework and Control Scheme

### 2.1. DyPLS Modeling

PLS was first proposed by Herman Wold's original non-linear iterative partial least square (NIPALS) algorithm [21]. The principle of PLS comprises two related models, outer model and inner model. Consider there are l scaled samples of input dataset $X$ with $m$ dimensions and output dataset $Y$ with $n$ dimensions. The correlation model of $X$ and $Y$ are obtained by the outer model, shown in Equation (1).

$$X = \sum_{r=1}^{R} t_r p_r^T + E^* = TP^T + E^*$$

$$Y = \sum_{r=1}^{R} u_r q_r^T + F^* = UQ^T + F^*$$

$$(1)$$

where, $T = [t_1, t_2, \cdots, t_R]$ and $U = [u_1, u_2, \cdots, u_R]$ are the score matrices of $X$ and $Y$, respectively. $P = [p_1, p_2, \cdots, p_R]^T$ and $Q = [q_1, q_2, \cdots, q_R]^T$ are the loading matrices of $X$ and $Y$, respectively. $E^*$ and $F^*$ are residual matrices of $X$ and $Y$, respectively. $R$ is the number of latent variables. By the outer model, $m$-dimension dataset $X$ and $n$-dimension $Y$ are mapped onto a lower $R$-dimension space. Shown in Equation (2) is the inner model of PLS. This constructs the relationship between score matrices $T$ and $U$. The diagonal coefficient matrix $B$ can be calculated by the least squares method. According to the principle of this model, the multivariate regression problem is decomposed into $R$ single-variable regression problems. In addition, $m \neq n$ is met in most cases; this method can also deal with non-square regression problems.

$$u_r = b_r t_r$$

$$U = TB$$

$$(2)$$

where $b_r = \frac{u_r^T t_r}{t_r^T t_r}$; $B = (T^T T)^{-1} T^T U$;. Combing Equation (1) with Equation (2), the PLS regression model can be written as

$$Y = TBQ^T + F^* = \sum_{r=1}^{R} b_r t_r q_r^T + F^*$$

$$(3)$$

A more detailed PLS algorithm has been presented elsewhere [22]. The conventional PLS is suitable for pure algebraic structures. It is not able to cope with dynamic characteristics in a process system. Researchers have proposed many different DyPLS models by incorporating structures like time-series terms or dynamic filters into the PLS structure [12,23]. In this paper, an ARX model is applied to the inner model of PLS [24] to represent the dynamic character of the process. This can be expressed as follows:

$$u_r(k) = A_r(q^{-1}) u_r(k-1) + B_r(q^{-1}) t_r(k) + \xi_r(k)$$

$$= [u_r(k-1), \cdots, u_r(k-n_a), t_r(k-1), \cdots, t_r(k-n_b)][a_{r,1}, \cdots, a_{r,n_a}, b_{r,1}, \cdots, b_{r,n_b}]^T + \xi_r(k) \qquad (4)$$

$$= \varphi_r(k)\theta_r^T + \xi_r(k) H_r(\varphi_r)$$

where, $A_r(q^{-1}) = -a_{r,1} - a_{r,2}q^{-1} - a_{r,n_a}q^{-n_a+1}$, $B_r(q^{-1}) = b_{r,1}q^{-1} + b_{r,2}q^{-2} \cdots + b_{r,n_b}q^{-n_b}$, $q^{-1}$ is the backward shift operator. $n_a$ and $n_b$ are the number of lags; $\xi_r(k)$ is the model error; $\theta_r^T = [a_{r,1}, \cdots, a_{r,n_a}, b_{r,1}, \cdots, b_{r,n_b}]^T$ is the parameter vector to be estimated; $\varphi_r(k)$ is the regressor vector. By prefiltering by the scores $t_r$ and $u_r$, ARMAX or CARIMA models can be reformed into an ARX form, which would improve the quality of Equation (4) and the closed loop performance based on it [25,26]. Bringing Equation (4) into Equation (3), the DyPLS model can be written as

$$Y(k) = \sum_{r=1}^{R} u_r(k) q_r^T + F^*(k) = U(k)Q^T + F^*(k)$$

$$(5)$$

where $U(k) = \text{diag}(u_1(k), u_2(k), \cdots, u_R(k))$.

### 2.2. Controller Design in the DyPLS Framework

The controller design in latent variable space proposed by Kaspar and Ray [12] is shown in Figure 1. Unlike conventional control methods, the controllers $G_c$ for the controlled system $G_p$ are designed under latent variable space, based on inner dynamic models (Equation (4)). In this framework, setpoint $Y$ set and system output $Y$ are scaled by scaling matrix $W_y^{-1}$ and mapped into latent variable space by inverse loading matrix $Q$. The control law $T$ is back mapped into original space by loading

matrix P and anti-scaled by matrix $W_x$; It is the same as the input of actual system. $D$ is the disturbance sequence, which also needs to be mapped into latent variable space. According to Equation (4), $G_c$ is $R$ single-input single-output (SISO) controllers. It inherits the features of PLS, and decouples multiple-input multiple-output (MIMO) systems into a series of SISO subsystems.

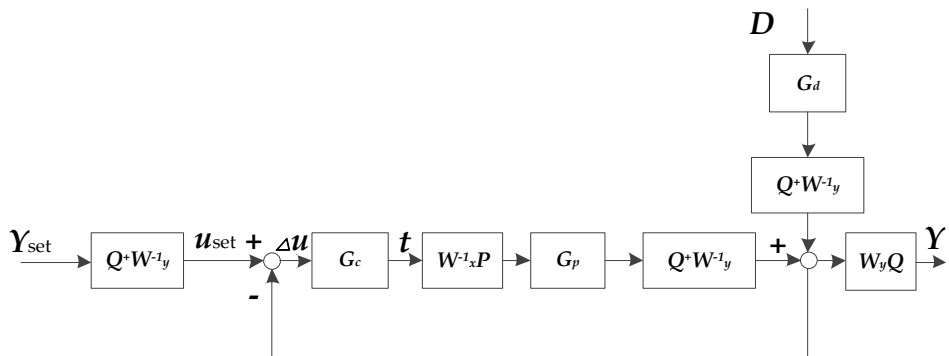

**Figure 1.** PLS control framework proposed by Kaspar and Ray.

## 3. Offset-Free Model Predictive Control in the DyPLS Framework

### 3.1. State Space-Based MPC in the DyPLS Framework

In some articles, PLS is called a latent subspace project method [27]. This is different from a subspace identification method (SIM). In a MIMO system, SIM is the method that identifies the state space model in original space; subsequently, many control algorithms can be applied to it. It does not map original space into latent variable space or decompose the MIMO system into multiple SISO subsystems. Hence, state space-based MPCs in the DyPLS are not suitable for SIM.

To simplify the description, it is assumed that the model error $\xi_r(k)$ in Equation (4) is zero, and $n_a$ is equal to $n_b$. The ARX model in the $r$-th latent variable space (Equation (4)) may be realized as a state space model in an innovation form [5]

$$
\begin{aligned}
x_r(k+1) &= A_r x_r(k) + B_r t_r(k) \\
u_r(k) &= C_r x_r(k)
\end{aligned}
\tag{6}
$$

where, the matrices $(A_r, B_r, C_r)$ having a canonical observer form,

$$
A_r = \begin{bmatrix} -a_{r,1} & 1 & 0 & \cdots & 0 \\ -a_{r,2} & 0 & 1 & \cdots & 0 \\ \vdots & \vdots & \vdots & \ddots & \vdots \\ -a_{r,n_a-1} & 0 & 0 & \cdots & 1 \\ -a_{r,n_a} & 0 & 0 & \cdots & 0 \end{bmatrix}, \ B_r = \begin{bmatrix} b_{r,1} \\ \vdots \\ b_{r,n_a} \end{bmatrix}, \ C_r = \begin{bmatrix} 1 \\ 0 \\ \vdots \\ 0 \end{bmatrix}^T.
$$

Let $N_{p,r}$ and $N_{c,r}$ denote prediction horizon and control horizon, respectively. The $N_{p,r}$ step ahead prediction of the output is $Y$

$$
\hat{u}_r(k) = \Psi_r x_r(k) + Y_r t_r(k-1) + \Theta_r \Delta \hat{t}_r(k)
\tag{7}
$$

where $\hat{u}_r(k) = \begin{bmatrix} \hat{u}_r(k+1|k) \\ \vdots \\ \hat{u}_r(k+N_{c,r}|k) \\ \hat{u}_r(k+N_{c,r}+1|k) \\ \vdots \\ \hat{u}_r(k+N_{p,r}|k) \end{bmatrix}$, $\mathbf{\Psi}_r = \begin{bmatrix} C_r A_r \\ \vdots \\ C_r A_r^{N_{c,r}} \\ C_r A_r^{N_{c,r}+1} \\ \vdots \\ C_r A_r^{N_{p,r}} \end{bmatrix}$, $Y_r = \begin{bmatrix} C_r B_r \\ \vdots \\ \sum_{i=0}^{N_{c,r}-1} C_r A_r^i B_r \\ \sum_{i=0}^{N_{c,r}} C_r A_r^i B_r \\ \vdots \\ \sum_{i=0}^{N_{p,r}} C_r A_r^i B \end{bmatrix}$,

$$\mathbf{\Theta}_r = \begin{bmatrix} C_r B_r & \cdots & 0 \\ C_r A_r B_r + C_r B_r & \cdots & 0 \\ \cdots & \cdots & \cdots \\ \sum_{i=0}^{N_{c,r}-1} C_r A_r^i B_r & \cdots & C_r B_r \\ \sum_{i=0}^{N_{c,r}} C_r A_r^i B_r & \cdots & C_r A_r B_r + C_r B_r \\ \cdots & \cdots & \cdots \\ \sum_{i=0}^{N_{p,r}-1} C_r A_r^i B_r & \cdots & \sum_{i=0}^{N_{p,r}-N_{c,r}} C_r A_r^i B_r \end{bmatrix}, \Delta\hat{t}_r(k) = \begin{bmatrix} \Delta\hat{t}_r(k|k) \\ \vdots \\ \Delta\hat{t}_r(k+N_{p,r}-1|k) \end{bmatrix}, \Delta = 1-z^{-1}.$$

A typical cost function of the 2-norm form used here is

$$J_r = \sum_{j=1}^{N_{p,r}} \delta_r(j)\|u_{set,r}(k+j)-\hat{u}_r(k+j|k)\|_2^2 + \sum_{j=0}^{N_{c,r}-1} \lambda_r(j)\|\Delta\hat{t}_r(k+j|k)\|_2^2 \tag{8}$$

where, $u_{set,r}$ is the setpoint in the latent space which is transformed from the setpoint in original space; $\delta_r(j)$ and $\lambda_r(j)$ are weighting sequences. Bring Equation (8) into Equation (7), and solving the minimization problem $J_r$, the following optimal set of future increment score matrices $\Delta t_r$ is obtained as

$$\Delta\hat{t}_r(k) = (\mathbf{\Theta}_r^T \delta_r \mathbf{\Theta}_r + \lambda_r)^{-1} \mathbf{\Theta}_r^T \delta_r [u_{set,r}(k) - \mathbf{\Psi}_r x_r(k) - Y_r t_r(k-1)] \tag{9}$$

where $\delta_r = diag(\delta_r(1), \cdots, \delta_r(N_{p,r}))$, $\lambda_r = diag(\lambda_r(0), \cdots, \lambda_r(N_{c,r}-1))$, $u_{set,r}(k) = [u_{set,r}(k+1), \cdots, u_{set,r}(k+N_{p,r})]^T$.

Only the first part of the solution $\Delta\hat{t}_r$ is back-mapped to the original space and implemented with respect to the process. One can guarantee closed-loop stability by choosing a sufficiently long prediction and control horizon. As usual, the input of plant model $G_p$ in Figure 1 is in the terms of input data matrix $X$ (in Equation (1)). There are two ways of mapping score moving $\Delta\hat{t}_r$ to the original space in order to get the manipulated variable $X$. One way is to integrate score $t_r$ with

$$t_r(k) = t_r(k-1) + \Delta\hat{t}_r(k|k) \tag{10}$$

and then to back map to original space with Equation (1). The other way is back map the score move $\Delta\hat{t}_r(k|k)$ into original space with

$$\Delta X(k) = \sum_{r=1}^{R} \Delta\hat{t}_r(k|k)p_r^T \tag{11}$$

and then to integrate the input data as

$$X(k) = X(k-1) + \Delta X(k) \tag{12}$$

From Equation (7), one can conclude that these decomposed SISO control problems are independent from each other, and latent variables are selected in pairs automatically. Hence, the state space MPC in DyPLS not only avoids decoupling the MIMO system, but also avoids pairing the control loop. All latent variables in cost function Equation (8) are solved separately. In addition, the computational complexity of this method is less than the MIMO control problem.

The structure of the control scheme is illustrated in Figure 2. The feedback of the control is the model state $x_r$. When the state space model in the DyPLS framework matches the plant model accurately, $x_r$ can explain the system output $Y$ well. Due to the decoupling scheme of PLS, the mismatch error $F^*(k)$ (in Equation (6)) is unavoidably present. This leads to poor control performance or steady state error. In the next two sections, two methods are proposed to solve this problem.

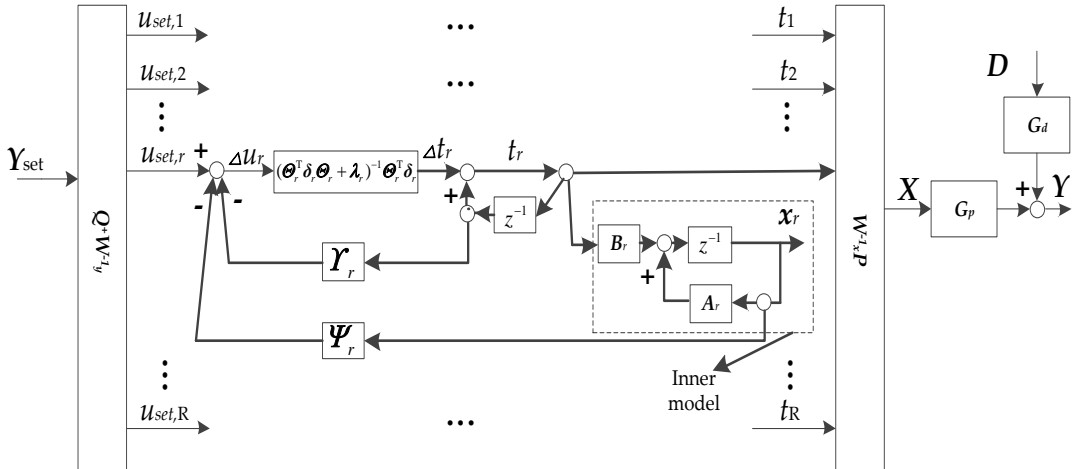

**Figure 2.** Structure of state space MPC control scheme in the PLS framework.

### 3.2. Offset-free MPC Method A

Since the cost function Equation (8) includes the score move $\Delta t_r$, rather than the score, $t_r$, itself, it is advantageous to reformulate the state model Equation (6), which has $\Delta t_r$ as the model input. From Equation (6), the movement of the state model at time $k$ can be formulated as

$$
\begin{aligned}
\Delta x_r(k+1) &= A_r \Delta x_r(k) + B_r \Delta t_r(k) \\
\Delta u_r(k) &= C_r \Delta x_r(k)
\end{aligned}
\tag{13}
$$

Then Equation (13) can be reformulated in an augmented form as

$$
\begin{bmatrix} \Delta x_r(k+1) \\ u_r(k+1) \end{bmatrix} = \begin{bmatrix} A_r & 0 \\ C_r A_r & 1 \end{bmatrix} \begin{bmatrix} \Delta x_r(k) \\ u_r(k) \end{bmatrix} + \begin{bmatrix} B_r \\ C_r B_r \end{bmatrix} \Delta t_r(k)
$$

$$
u_r(k) = \begin{bmatrix} 0 & 1 \end{bmatrix} \begin{bmatrix} \Delta x_r(k) \\ u_r(k) \end{bmatrix}
\tag{14}
$$

Equation (14) can be rewritten in a reduced form as

$$
\begin{aligned}
Z_r'(k+1) &= A_r' Z'(k) + B_r' \Delta t_r(k) \\
u_r(k) &= C_r' Z_r'(k)
\end{aligned}
\tag{15}
$$

where $Z_r'(k) = \begin{bmatrix} \Delta x_r(k) \\ u_r(k) \end{bmatrix}$, $A_r' = \begin{bmatrix} A_r' & 0 \\ C_r A_r' & 1 \end{bmatrix}$, $B_r' = \begin{bmatrix} B_r' \\ C_r B_r' \end{bmatrix}$, $C_r' = \begin{bmatrix} 0 & 1 \end{bmatrix}$.

The $N_{p,r}$ step ahead prediction of the output for $j = 1, \ldots, N_{p,r}$ is

$$\hat{\boldsymbol{u}}_r(k) = \boldsymbol{\Theta}_r \boldsymbol{Z}'(k) + \boldsymbol{\Gamma}_r \Delta \hat{\boldsymbol{t}}_r(k) = \boldsymbol{\Theta}_r \boldsymbol{x}_r(k) + \boldsymbol{u}_r(k) + \boldsymbol{\Gamma}_r \Delta \hat{\boldsymbol{t}}_r(k) \tag{16}$$

$$\text{where, } \boldsymbol{\Theta}_r = \begin{bmatrix} C_r' A_r' \\ C_r' A_r'^2 \\ \cdots \\ C_r' A_r'^{N_{c,r}} \\ C_r' A_r'^{N_{c,r}+1} \\ \cdots \\ C_r' A_r'^{N_{p,r}} \end{bmatrix}, \boldsymbol{\Theta}_r' = \begin{bmatrix} C_r A_r \\ C_r A_r^2 + C_r A_r \\ \cdots \\ \sum\limits_{j=1}^{N_{c,r}} C_r A_r^j \\ \sum\limits_{j=1}^{N_{c,r}+1} C_r A_r^j \\ \cdots \\ \sum\limits_{j=1}^{N_{p,r}} C_r A_r^j \end{bmatrix},$$

$$\boldsymbol{\Gamma}_r = \begin{bmatrix} C_r' B_r' & 0 \\ C_r' A_r' B_r' & C_r' B_r' & \ddots \\ \cdots & & \ddots & 0 \\ C_r' A_r'^{N_{c,r}-1} B_r' & & & C_r' B_r' & 0 \\ C_r' A_r'^{N_{c,r}} B_r' & & & C_r'(A_r' + I)B_r' & 0 \\ \cdots & \cdots & \cdots & \cdots & & \ddots \\ C_r' A_r'^{N_{p,r}-1} B_r' & \cdots & \cdots & C_r' \sum\limits_{j=0}^{N_{p,r}-N_{c,r}} A_r'^j B_r' & & 0 \end{bmatrix}$$

By solving the cost function of Equation (8), the following optimal set of future increment score matrices $\Delta \hat{\boldsymbol{t}}_r$ is obtained:

$$\Delta \hat{\boldsymbol{t}}_r(k) = \left( \boldsymbol{\Gamma}_r^T \delta_r \boldsymbol{\Gamma}_r + \lambda_r \right)^{-1} \boldsymbol{\Gamma}_r^T \delta_r (r(k) - \boldsymbol{\Phi}_r' \Delta \boldsymbol{x}_r(k) - \boldsymbol{u}_r(k)) \tag{17}$$

The control structure is illustrated in Figure 3. When the DyPLS model matches the plant precisely, the system output score $u_{Y,r}(k)$ is equal to the predictive output of the state space model in the latent space $u_r(k)$. As described above, the DyPLS method makes a tradeoff between the complexity and dimensions of the model structure and the accuracy of the model. Chi et al. [28] point out that this tradeoff would lead to a small degradation of control performance. That is to say, $u_r(k)$ will not precisely map the true value of the system output. Hence, one can replace $u_r(k)$ with $u_{Y,r}(k)$, and the system output mapped in latent variable space could be used as the feedback of the control scheme. Based on the stability and convergence of MPC, the control performance will be improved. In this study, DyPLS model is used to decompose the MIMO system into several SISO subsystems, so that a SISO velocity form state space MPC is available. The stability and convergence analysis of the velocity form state space MPC in a SISO case has been illustrated by Liuping [29] and Betti [30].

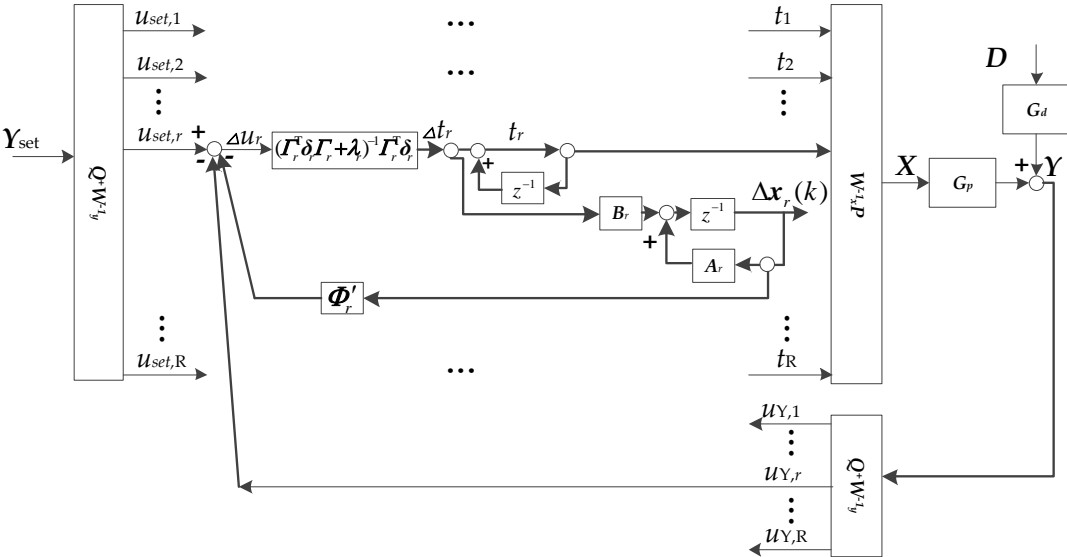

**Figure 3.** The control structure of the move form state mode.

*3.3. Offset-free MPC Method B*

The other method to eliminate the offset is to apply a state observer, which is usually incorporated with a disturbance model. In this section, the state space model is augmented with a disturbance model, which is used to eliminate unmeasured disturbance, and an observer is introduced to guarantee the offset-free tracking performance.

As shown in Figure 1, the disturbances acting on the plant are the output disturbance. At each time instant, the current and future disturbances are usually unknown. It is difficult to measure the disturbance, especially for unmeasured disturbance. In most cases, it is assumed that the disturbance will be unchanged during the prediction horizon. Hence, the inner model Equation (6) is augmented with a disturbance form as

$$\begin{aligned} x_r(k+1) &= A_r x_r(k) + B_r t_r(k) \\ u_r(k) &= C_r x_r(k) + d_r(k) \end{aligned} \tag{18}$$

With Equation (7), the $N_{p,r}$ steps ahead prediction of the output is rewritten as

$$\hat{u}_r(k) = \boldsymbol{\Psi}_r x_r(k) + Y_r t_r(k-1) + \boldsymbol{\Theta}_r \Delta \hat{t}_r(k) + \hat{d}_r(k) \tag{19}$$

Where, $\hat{d}_r(k) = \begin{bmatrix} \hat{d}_r(k|k) \\ \vdots \\ \hat{d}_r(k+N_{p,r}-1|k) \end{bmatrix} = \begin{bmatrix} \hat{d}_r(k|k) \\ \vdots \\ \hat{d}_r(k|k) \end{bmatrix} \hat{d}_r(k|k) = u_{Y,r}(k) - \hat{u}_r(k|k-1).$

Since Equation (6) cannot measure the full state of the plant, an observer is used to estimate the state vector [31]. The condition for the observability of Equation (18) is given in the following proposition, which is extended from Maeder's results [32].

**Proposition 1.** *The augmented state space model Equation (18) is observable if and only if $(A_r, C_r)$ is observable and*

$$\begin{bmatrix} A_r - \lambda I & \mathbf{0} \\ C_r & 1 \end{bmatrix} \tag{20}$$

*has full column rank.*

**Proof.** From the Hautus observability condition augmented system, (18) is observable if and only if

$$\begin{bmatrix} A_r^T - I & 0 & C_r^T \\ 0 & 1 - \lambda & 1 \end{bmatrix}$$ has full row rank $\forall \lambda$.

The first set of rows is linearly independent if and only if $(A_r, C_r)$ is observable. The second set of rows is linearly independent of the first set of rows, except for possibly $\lambda = 1$. Thus, for the augmented system, the Hautus condition needs to be checked for $\lambda = 1$ only. This means that Equation (20) should be met. □

The state observer designed based on Equation (18) is

$$x_r(k+1) = A_r x_r(k) + B_r t_r(k) + L_r'(u_{Y,r}(k) - \hat{u}_r(k|k-1)) \tag{21}$$

where, $L_r'$ is the gain matrix to estimate the correct state $x_r$. By appropriate design of $L_r'$, the state estimator can facilitate offset-free control.

There are many methods to obtain an appropriate gain matrix $L_r'$, such as the state-feedback pole-placement method. Meanwhile, if the state and output equations of the plant are assumed to be subjected to white noise disturbances with known covariance matrices, $L_r'$ can be obtained by Kalman Filters [25]. For application of the Kalman Filter, the underlying requirement is the disturbance covariance, which is used in calculation of the estimator gain. This is estimated from the auto-covariance of the plant data. In the DyPLS model, covariance of plant data is mapped into latent variable space. The characteristic of covariance of it is shown as Proposition 2.

**Proposition 2.** *Let $\sigma_r$ denote the variance vector of the r-th score vector in the latent variable space. $\sigma_r$ is the sum of r-th vector of $Q^+$.*

**Proof.** According to Equation (1), the score variable $U$ is obtained by $U = YQ^+$ and $u_r = [y_1, \cdots, y_n] \cdot q_r^+$, where, $q_r^+$ is the column vector of $Q^+$. Based on DyPLS scheme, the output data set are scaled to the unit variance data set. $y_1, \cdots, y_n$ all belong to the unit variance data set. Assuming that all the system outputs are independent of each other, based on the property of variance, $\sigma_r$ is the sum of element of $q_r^+$. □

Based on Equation (19), the optimal future increment of score vector $\Delta t_r$ can be obtained, as

$$\Delta \hat{t}_r(k) = (\Theta_r^T \delta_r \Theta_r + \lambda_r)^{-1} \Theta_r^T \delta_r [u_{set,r}(k) - \Psi_r x_r(k) - Y_r t_r(k-1) - \hat{d}_r(k)] \tag{22}$$

Replace the state variable $x_r$ in Equation (22) with the state variable in Equation (21), and one can get the future increment of score vector $\Delta t_r$ with offset-free tracking performance. The structure of the *r*-th sub controller is shown in Figure 4.

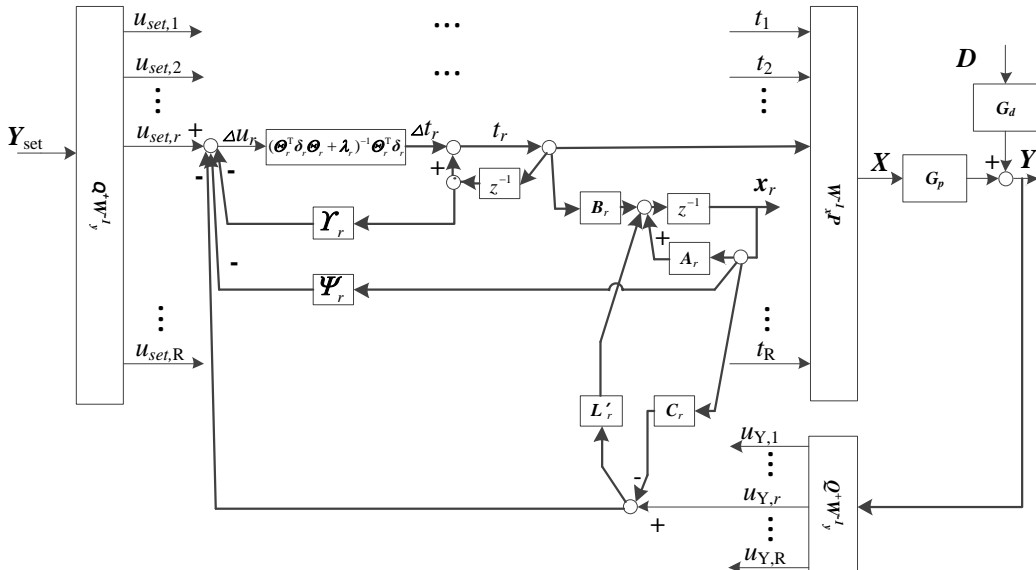

**Figure 4.** The control structure with disturbance model and observer.

## 4. Case Study

To illustrate the performance of the proposed offset-free MPC based on the dynamic PLS framework, two study cases are presented below.

### 4.1. Study Case 1: Jerome-Ray Distillation Column

The Jerome-Ray distillation column is a non-minimum phase system, which has zero points at right half-plane [33]. A lot of multivariate control algorithms have been proposed to achieve a better control performance. In this case study, the basic state space-based MPC (SMPC), state space-based MPC in DyPLS (SMDP), and two offset-free state space-based MPCs in DyPLS (OSMDP1, OSMDP2) are illustrated to compare their control performance. The transfer function matrix of this process is given as

$$G(s) = \begin{bmatrix} \frac{(-s+1)e^{-2s}}{s^2+1.5s+1} & \frac{0.5(-s+1)e^{-4s}}{(2s+1)(3s+1)} \\ \frac{0.33(-s+1)e^{-6s}}{(4s+1)(5s+1)} & \frac{(-s+1)e^{-3s}}{4s^2+6s+1} \end{bmatrix} \tag{23}$$

To simulate the disturbance of the real process, a disturbance model [34] is added as

$$G_d(s) = \begin{bmatrix} \frac{e^{-s}}{(25s+1)} & \frac{e^{-s}}{(25s+1)} \end{bmatrix}^T \tag{24}$$

In this simulation, the sampling time is set to 0.5 s. To build up the DyPLS model, two random step input signals with magnitudes ranging from −1 to 1 are applied to excite the system. White noise with a non-zero deviation of 0.5 is added to $G_d$ as a disturbance signal, which is used to simulate the real process situation. The input and output data to excite the system are plotted in Figure 5. In addition, another dataset, which is used to verify the model accuracy, is shown in Figure 6.

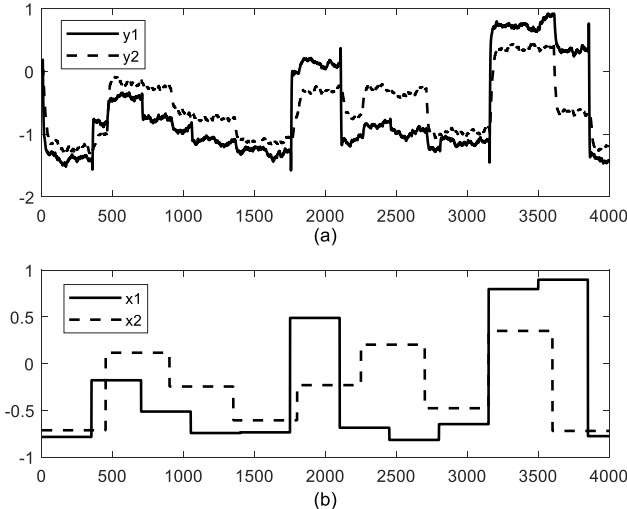

**Figure 5.** Input and output data to excite the system: (**a**) output data; (**b**) input data.

To determine the number of latent variables, the performance of the model is quantified by an indicator as:

$$\phi = \sum_{i=1}^{n} \sum_{k=1}^{l} (y_i(k) - \hat{y}_i(k))^2$$

where $y_i(k)$ and $\hat{y}_i(k)$ denote modeling data and DyLS output, respectively. $\phi$ reflects the bias between DyPLS output and modeling data. The $\phi$ for different model parameters are shown in Table 1. $\phi$ for $r = 2$ is significantly less than that for $r = 1$. Hence, the latent variable in this case is 2. For the model order, $n_a$ and $n_b$ are 5 and 5, because higher model orders do not significantly reduce the index $\phi$. In addition, higher model orders lead to heavier computational burden. To SMPC, the model order is 28. One can see that each sub model in latent variable space is much simpler than the conventional model in original space.

**Table 1.** The index $\varphi$ for different model parameters.

| $(n_a, n_b)$ | (1,1) | (2,2) | (3,3) | (4,4) | (5,5) | (6,6) | (7,7) | (8,8) | (9,9) |
|---|---|---|---|---|---|---|---|---|---|
| $\varphi(r = 1)$ | 802.35 | 805.49 | 806.92 | 806.27 | 805.32 | 804.62 | 805.53 | 804.63 | 803.88 |
| $\varphi(r = 2)$ | 549.53 | 538.15 | 538.97 | 538.02 | 537.54 | 537.39 | 541.68 | 537.99 | 538.32 |

With the obtained dynamic PLS model, the proposed control algorithms and original MIMO state space MPC is designed to track the square-wave signal. A positive step of 2 units is introduced in the reference for $y_1$ at time 201 to 1400, and for $y_2$ at time 601 to 2000. The prediction horizon and control horizon for 4 control algorithms is shown in Table 2. The simulation results are shown in Figure 7. To compare the control performance with the disturbance situation, a white noise with a non-zero deviation of 0.5 is added to $G_d$ as a disturbance signal for all simulation runs; the results are shown in Figure 8.

**Table 2.** The parameters for 4 control algorithms and computing time.

|  | SMPC | SMDP | OSMDP1 | OSMDP2 |
|---|---|---|---|---|
| $N_p$ | 9 | 6 | 6 | 6 |
| $N_u$ | 5 | 5 | 5 | 5 |
| Computing time (ms) | 406.01 | 386.36 | 283.45 | 385.52 |

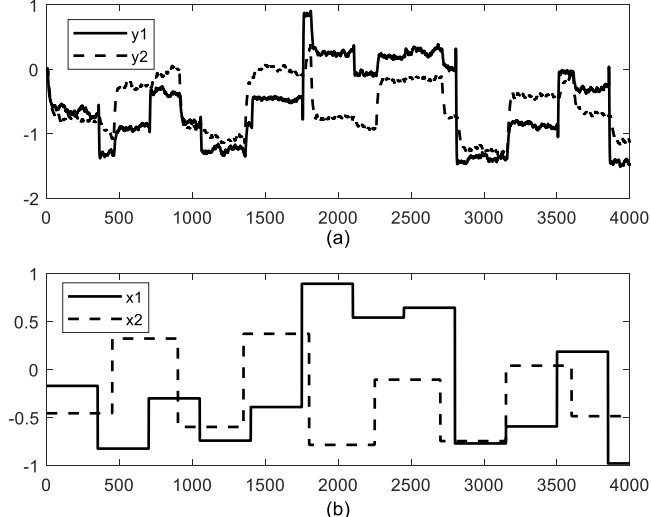

**Figure 6.** Input and output data to verify the system: (**a**) output data; (**b**) input data.

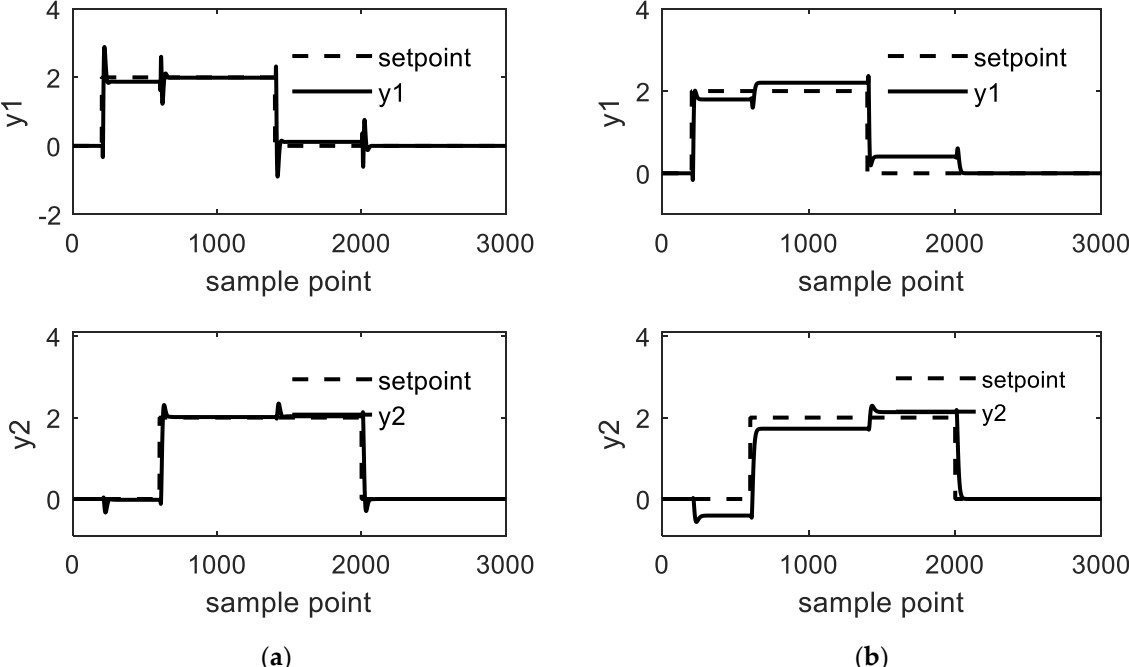

(**a**)                                                                                    (**b**)

**Figure 7.** *Cont.*

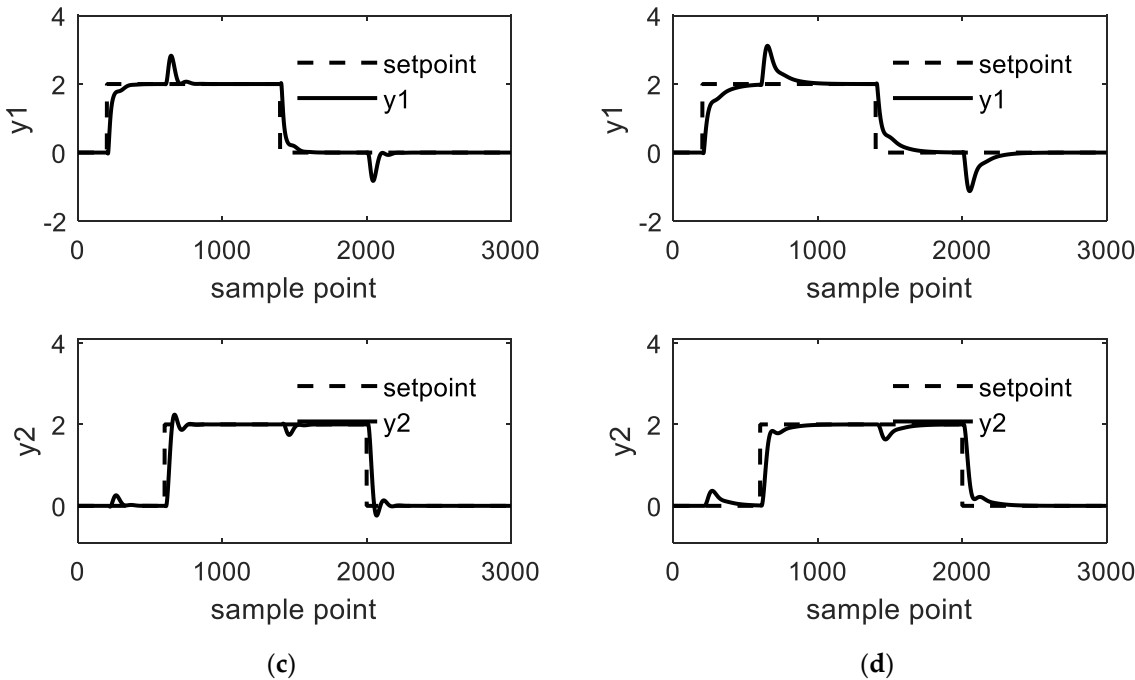

**Figure 7.** The response of the system without disturbance: (**a**) SMPC; (**b**) SMDP; (**c**) OSMDP1; (**d**) OSMDP2.

Due to the mismatch of the plant model and the predictive model, SMPC and SMDP achieve a poor offset-free performance. Although the latent variable in this case cannot be reduced, the decoupling scheme of dynamic PLS model makes a tradeoff between the control performance and the control structure complexity. That is to say, the steady state error of SMDP is larger than SMPC. OSMDP1 and OSMDP2 provide a good offset-free performance when there is no disturbance. OSMDP1 is sensitive to the colored noise. However, OSMDP2 is stable when noise exists. This indicates that OSMDP2 is more robust than OSMDP1.

The $N_p$ and Nu of SMPC are larger than for other methods. The identified model of SMPC has a large time delay. To include the dynamic of the system in the controller, the predictive horizon cannot be less than 9. The form of SMPC future control increment has the same form as Equation (9). In this case, $\Theta$ of SMPC is a $18 \times 10$ dimensional matrix, while for the other 3 methods it is two $6 \times 5$ dimensional matrices. Hence, the calculation complexity of matrix inverse and time consumption are high for SMPC. The last row of Table 2 is the computing time (the computer with 4 GB RAM, 2.6 GHz core i5) of 4 methods for the whole simulation. As can be seen, the computing time in the DyPLS framework is less than that in the original space MPC. The main reason for this decrease is that latent variable space controllers compute in parallel.

### 4.2. Study Case 2: Industrial Polyethylene Reaction

A common type of high-density polyethylene made by the catalytic homopolymerization of ethylene through slurry polymerization was proposed by Embirucu and Fontes [35]. The typical process has 9 inputs and 7 outputs. In this case, the first three equations shown in Equation (25) are extracted. These equations have 4 inputs ($x_1$—monomer feed flow, $x_2$—solvent (*n*-hexane) feed flow, $x_3$—catalyst feed flow, $x_4$—gas recycle/monomer feed ratio) and 3 outputs ($y_1$—production, $y_2$—slurry polymer, $y_3$—catalyst efficiency), which is a typical non-square system.

$$
\begin{aligned}
(1 - 0.9021q^{-1})y_1(k) &= (0.9283 - 0.8350q^{-1})x_1(k) \\
(1 - 0.9067q^{-1})y_2(k) &= (0.8415 - 0.7664q^{-1})x_1(k) + (0.6873 - 0.6023q^{-1})x_2(k) \\
(1 - 0.8932q^{-1})y_3(k) &= (0.8591 - 0.7536q^{-1})x_1(k) + (0.8097 - 0.7066q^{-1})x_3(k) + 0.0081x_4(k)
\end{aligned}
\tag{25}
$$

The square signals with a deviation of 0.01 white noise shown in Figure 9 were generated to excite the system. The corresponding output data are shown in Figure 10. The validation input data are shown in Figure 11. In addition, the output data of the dynamic PLS model is shown in Figure 12.

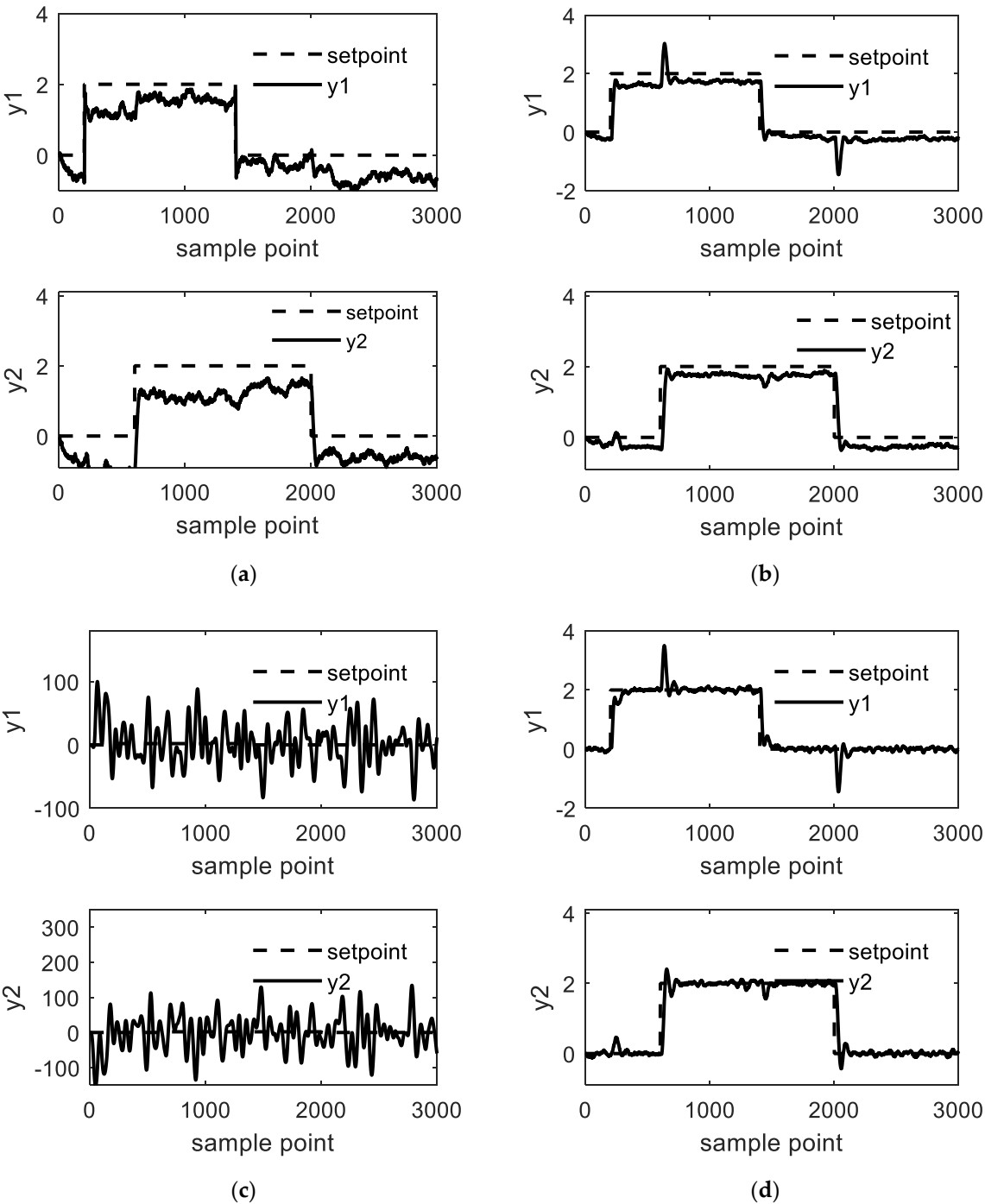

**Figure 8.** The response of system with disturbance (**a**) SMPC; (**b**) SMDP; (**c**) OSMDP1; (**d**) OSMDP2.

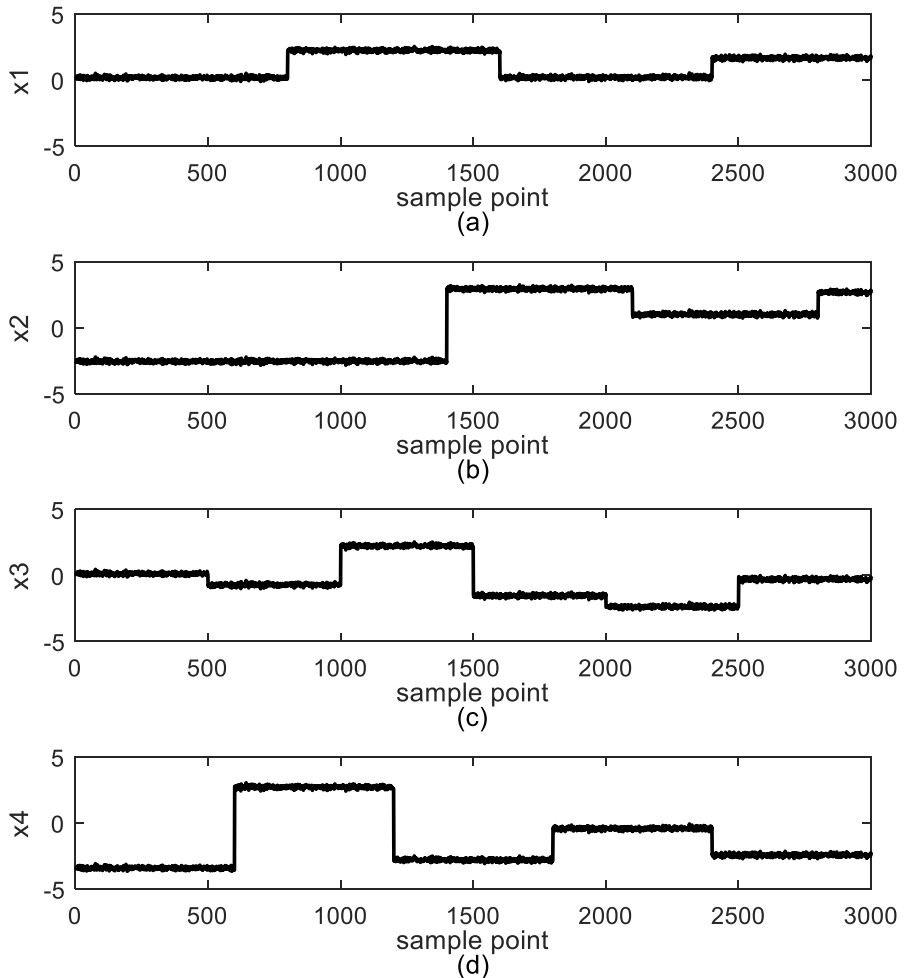

**Figure 9.** Input data set for polyethylene reaction: (**a**) $x_1$—monomer feed flow; (**b**) $x_2$—solvent (*n*-hexane) feed flow; (**c**) $x_3$—catalyst feed flow; (**d**) $x_4$—gas recycle/monomer feed ratio.

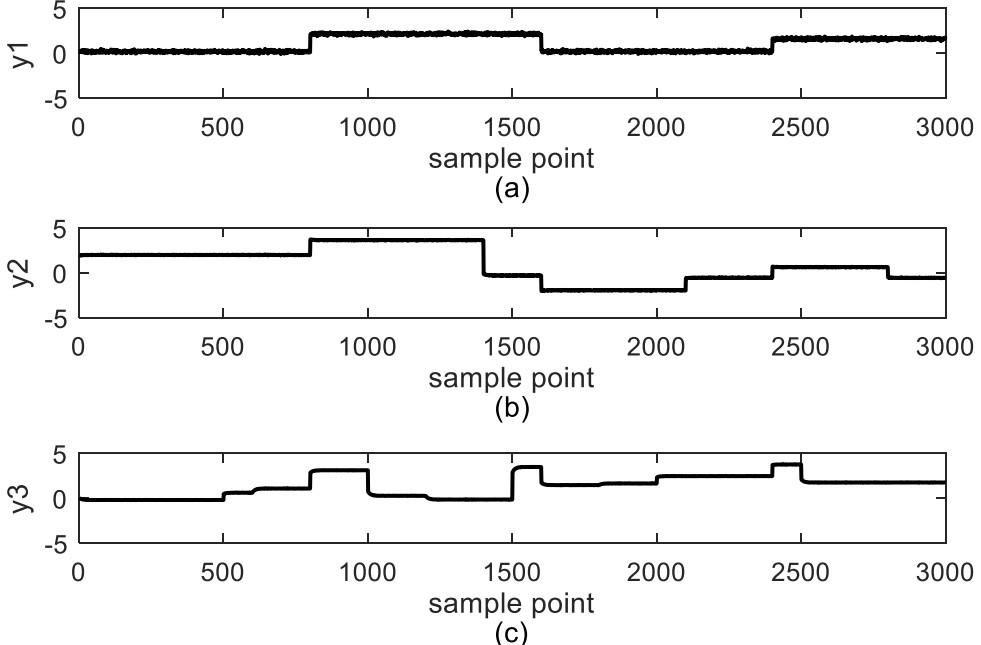

**Figure 10.** Output data set for polyethylene reaction: (**a**) $y_1$—production; (**b**) $y_2$—slurry polymer; (**c**) $y_3$—catalyst efficiency.

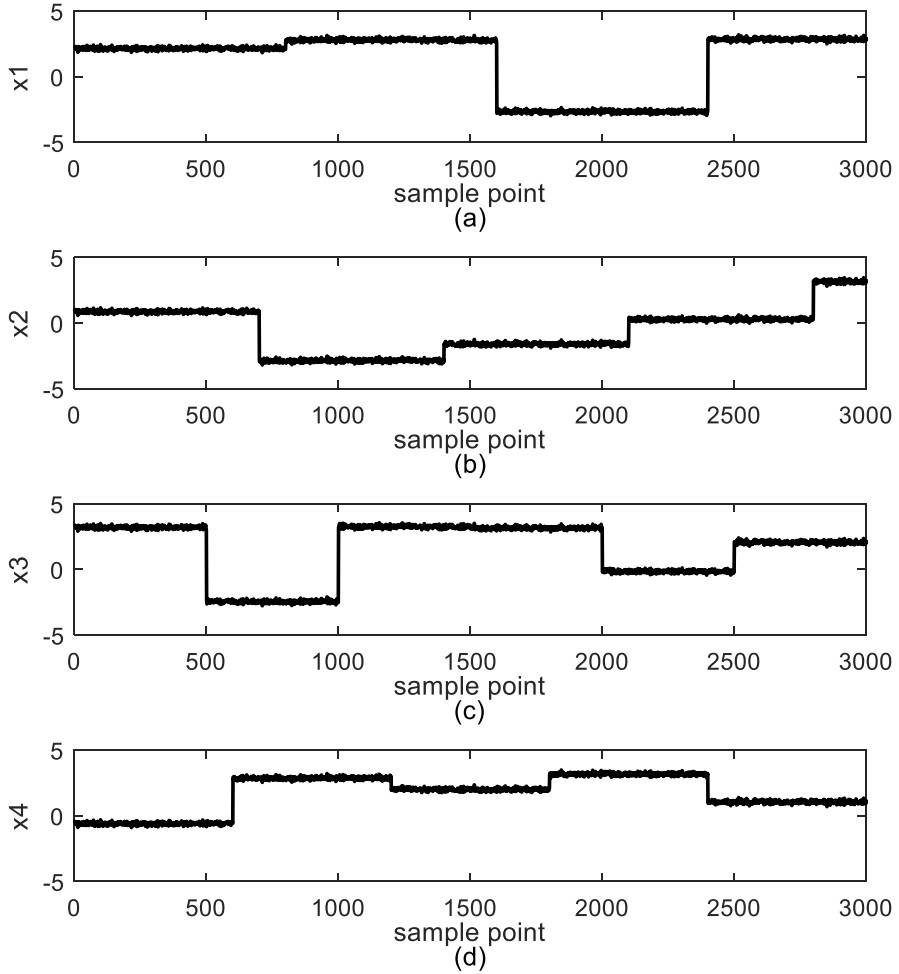

**Figure 11.** Validation input data of the system: (**a**) $x_1$—monomer feed flow; (**b**) $x_2$—solvent (*n*-hexane) feed flow; (**c**) $x_3$—catalyst feed flow; (**d**) $x_4$—gas recycle/monomer feed ratio.

The indicator $\phi$ of different latent variables and lagged parameters $n_a$, $n_b$ are illustrated in Table 3. The $\phi$ of 4 latent variables is significantly less than that of 2 or 3 latent variables. In addition, $\phi$ of $n_a = 4$ and $n_b = 4$ is less than other parameters. That is to say, the dynamic PLS model with 4 latent variables and $n_a = 4$, $n_b = 4$ can describe plant models well.

**Table 3.** The indicator $\phi$ for different latent variables and lagged parameters of ARX in dynamic PLS.

| Number of Latent Variable | $(n_a, n_b)$ | | | |
|---|---|---|---|---|
| | (2,2) | (4,4) | (6,6) | (8,8) |
| 2 | 8975.50 | 8976.50 | 8977.00 | 8961.8 |
| 3 | 1195.2 | 1193.5 | 1182.4 | 1187.6 |
| 4 | 126.88 | 114.67 | 128.3 | 130.08 |

The total simulation horizon for the control comparison is 1200. A unit signal is set as the setpoint of outputs. In addition, 3 unmeasured step disturbances (between 150 and 450 for $y_1$, between 550 and 750 for $y_2$, between 850 and 1050 for $y_3$) are acting on the 3 outputs of system. To compare the control performance when there is unmeasured white noise, a white noise is added to all outputs of the system. The simulation results with no noise are shown in Figures 13–15, and results with white noise are shown in Figures 16–18. The prediction and control horizon are 5 and 6 for these simulations, respectively. Results in Figures 13 and 16 show that SMDP is incapable of providing offset-free control, whether there is white noise or not. The two other control algorithms, OSMDP1 and OSMDP2, reject

these disturbances and provide offset-free tracking, see Figures 14, 15, 17 and 18. Results in Figures 14, 15, 17 and 18 also show that when unmeasured disturbance enter the system, OSMDP1 and OSMDP2 provide a correction. When unmeasured disturbances enter the system at sample points $150(y_1)$, $550(y_2)$ and $850(y_3)$; there are overshoots in the system, but they are rapidly declined, because in the proposed method, the actual outputs are introduced to latent variable space as the feedback.

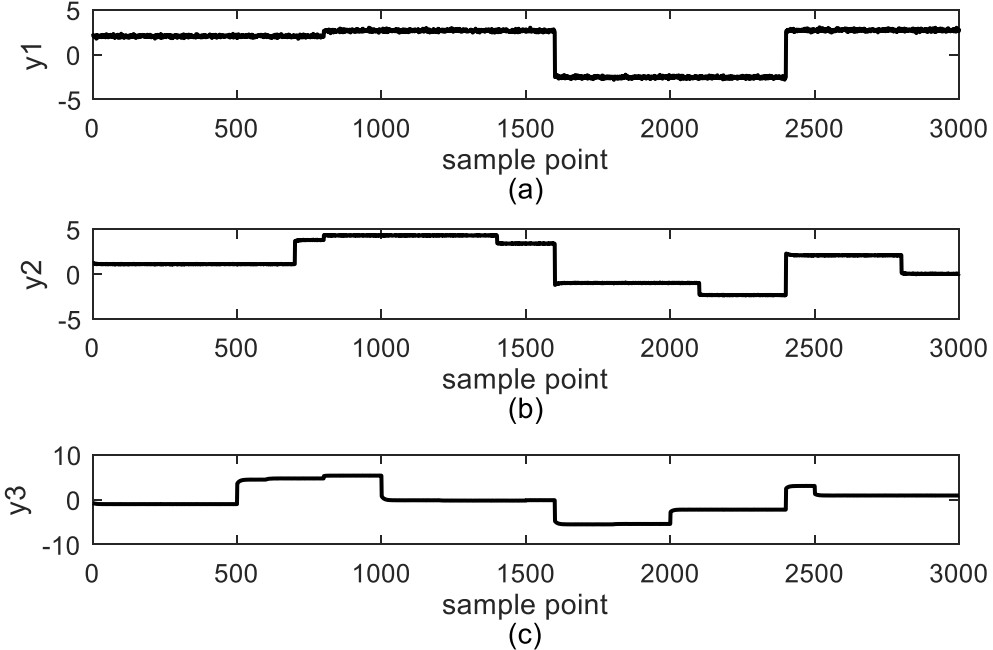

**Figure 12.** Output data of dynamic PLS model: (**a**) $y_1$—production; (**b**) $y_2$—slurry polymer; (**c**) $y_3$—catalyst efficiency.

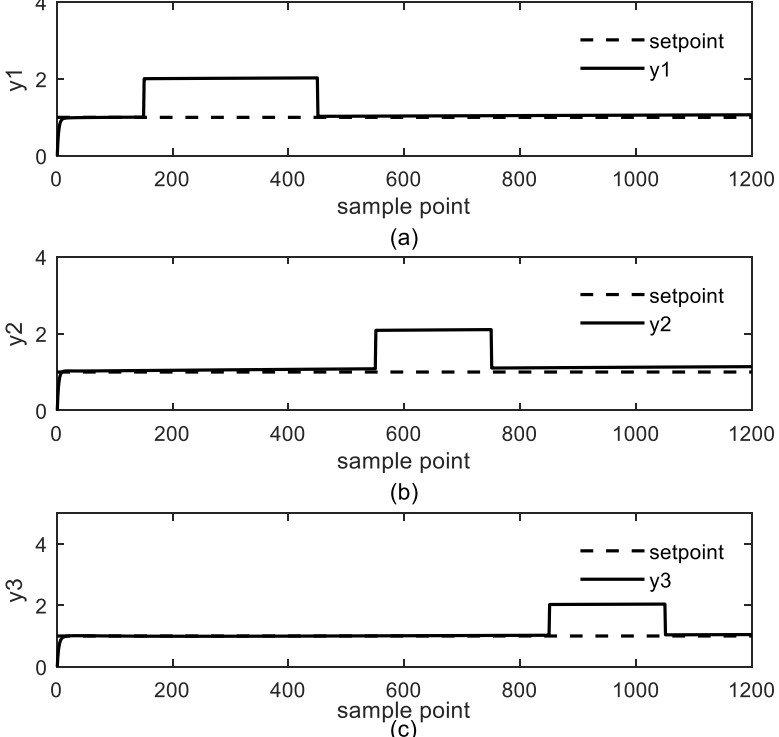

**Figure 13.** The simulation result of MSDP without noise: (**a**) $y_1$—production; (**b**) $y_2$—slurry polymer; (**c**) $y_3$—catalyst efficiency.

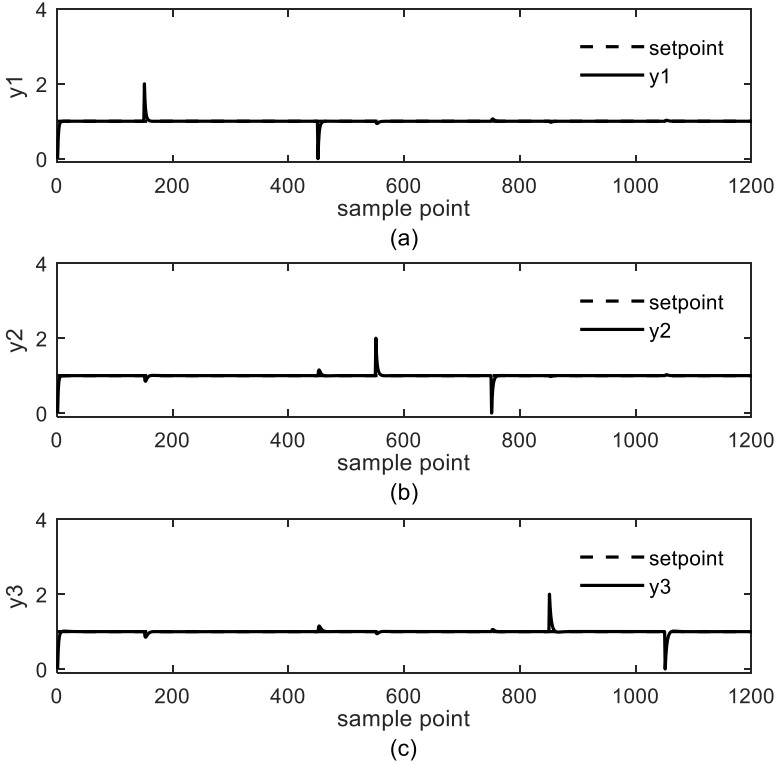

**Figure 14.** The simulation result of OSDP1 without noise: (**a**) $y_1$—production; (**b**) $y_2$—slurry polymer; (**c**) $y_3$—catalyst efficiency.

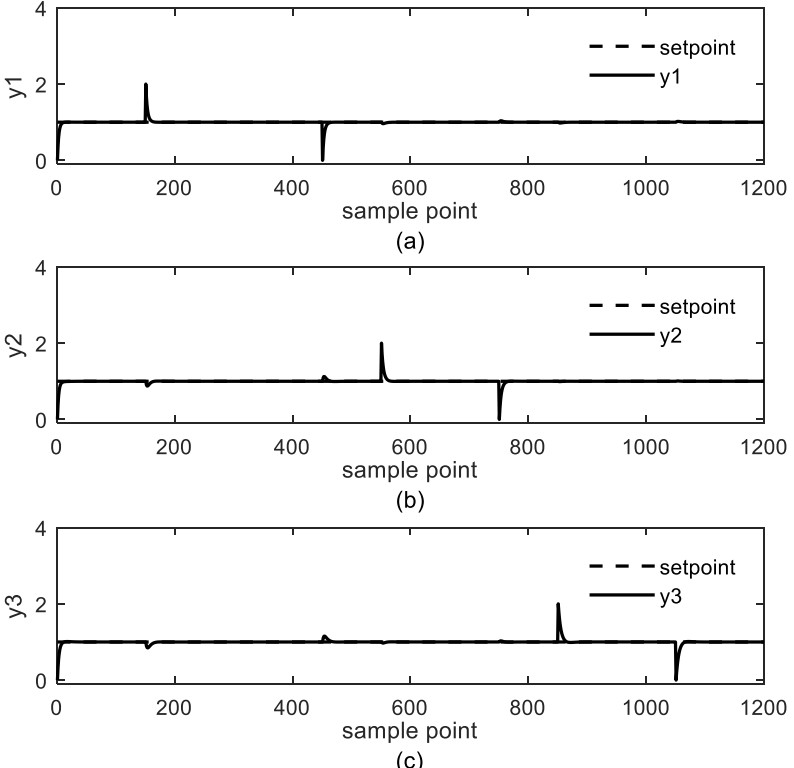

**Figure 15.** The simulation result of OSDP2 without noise: (**a**) $y_1$—production; (**b**) $y_2$—slurry polymer; (**c**) $y_3$—catalyst efficiency.

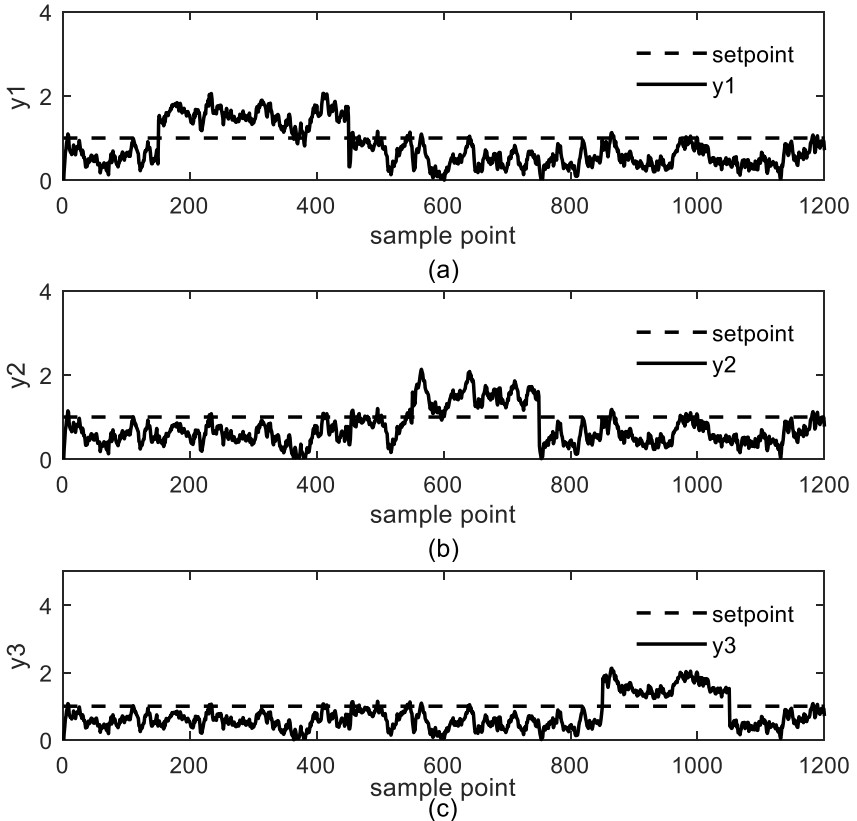

**Figure 16.** The simulation result of MSDP with white noise: (**a**) $y_1$—production; (**b**) $y_2$—slurry polymer; (**c**) $y_3$—catalyst efficiency.

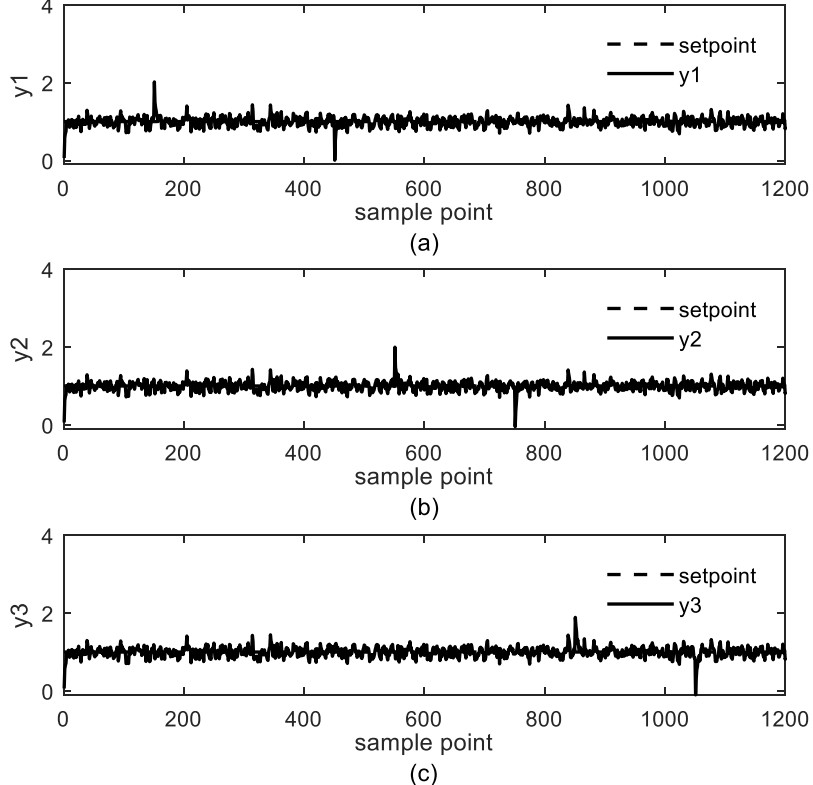

**Figure 17.** The simulation result of OSDP1 with white noise: (**a**) $y_1$—production; (**b**) $y_2$—slurry polymer; (**c**) $y_3$—catalyst efficiency.

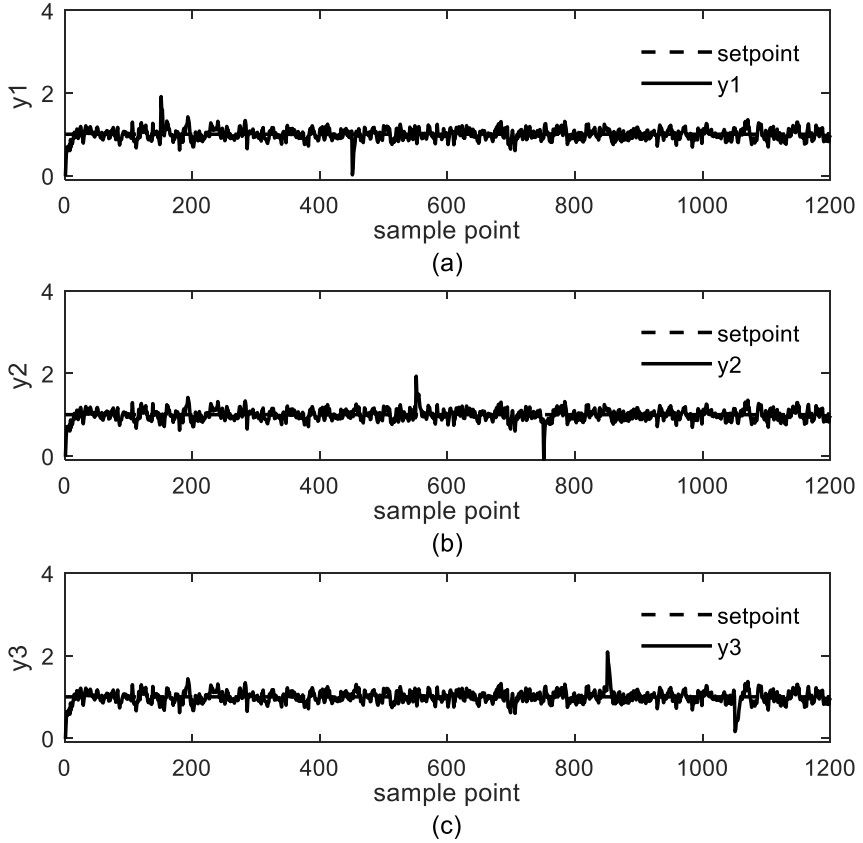

**Figure 18.** The simulation result of OSDP2 with white noise: (**a**) $y_1$—production; (**b**) $y_2$—slurry polymer; (**c**) $y_3$—catalyst efficiency.

## 5. Conclusions

In this paper, an MPC controller design in the DyPLS framework with an offset-free mechanism was proposed. First, the state space MPC in the DyPLS is proposed. This has five advantages: (1) the model structure is simple; (2) it decomposes the MIMO system into multiple SISO subsystems; (3) it can accomplish loop pairing automatically; (4) it can handle non-square systems; and (5) the dimensions of the system can be reduced. Meanwhile, due to the mismatch between the DyPLS model and the plant system, steady-state error exists. To tackle this problem, two methods are proposed. One is to reform the state model as a velocity form, where the state is composed by the state and output increments, while the manipulated variable is the control increment. The other is to augment the state space with a disturbance model, while assuming the disturbance model would be unchanged during the prediction horizon. In addition, an observer is used to estimate the unmeasured state. To obtain the Kalman filter gain matrix in the observer, the variance in the latent variable space is discussed. The second method shifts the focus from modeling the disturbance to estimation of the observer gain, which provides a significant simplification. A square system and a non-square system simulation illustrate the performance of proposed methods. Both offset-free methods can guarantee the offset-free tracking performance and unmeasured disturbance rejecting performance. Both methods can tackle the white noise in the measurement. However, the velocity form method is sensitive to colored noise.

**Author Contributions:** L.H.; writing—review and editing. Z.W.; case study. X.J.; methodology and writing. Y.W.; resources.

**Funding:** This study is financially supported by Scientific Research Fund of Liaoning Provincial Education Department (L2017LQN032, L2017LQN030) and Talent Scientific Research Fund of LSHU (2016XJJ-102).

**Conflicts of Interest:** The authors declare no conflict of interest.

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
