# Peer review of "Linear Offset-Free Model Predictive Control in the Dynamic PLS Framework"

_information, doi:10.3390/info10010005_

Round 1

Reviewer 1 Report

This paper proposes a linear offset-free MPC in the dynamic PLS framework with a numerical
simulation verification. This paper is well organized and easy to understand. Thus, the reviewer
recommends the acceptance for the publication, provided that few minor revisions are made
accordingly. The detailed comments on the manuscript are as follows.
1. A guideline for choosing the length of horizon should be included.
2. Why did not handle the input and state constraints? Please explain this.
3. More critical analysis of experimental results is required to show the effectiveness of the
proposed controller.
4. In all paragraph, the mathematical symbol must be distinguished with the text. Please see
and follow the sample manuscript format.
5. Please re-check the equation references.
6. Recent states of arts published in MDPI journals within three years have to be cited in the
introduction section.

Author Response

Dear reviewer,

 Thank you for your review. Our response for your comments are as follows. If you have any new comments,please contact us.  

Response to Reviewer 1 Comments

Comments and Suggestions for Authors

Point 1: A guideline for choosing the length of horizon should be included.

Response 1: the MPC in latent variable space is similar to the traditional MPC method. The rule for control horizon and predictive horizon is the same. So in the manuscript, we did not discuss these rules again.

Point 2: Why did not handle the input and state constraints? Please explain this.

Response 2: In reference[15], LÜ and Liang have handle the constraints in dynamic PLS framework already. So in the manuscript, we did not discuss it again.

Point 3: More critical analysis of experimental results is required to show the effectiveness of the proposed controller.

Response 3: authors improved the simulation discussion. Please see the revised.

Point 4: In all paragraph, the mathematical symbol must be distinguished with the text. Please see and follow the sample manuscript format.

Response 4: We are sorry for my poor mistake. And we corrected the mistake.

Point 5: Please re-check the equation references.

Response 5: We are sorry for my poor mistake. And we corrected the mistake.

Point 6: Recent states of arts published in MDPI journals within three years have to be cited in the introduction section.

Response 6: I have update introduction with 3 new references. Please see the revised.

Reviewer 2 Report

The present work designs a model predictive control (MPC) of offset free tracking problem in the dynamic partial least square (DyPLS) framework. And two methods are proposed to solve the offset free problem. Both are described in the manuscript, where the unmeasured step disturbance is rejected

The results are based on simulations that demonstrate the effectiveness of proposed methodology.

The manuscript must be corrected. for example In the text there are many broken links (Error! Reference source not found). In lines 80,85,92,93,104,105,106,117,130,131,141,142,149,151,154,158,163,169,171,172,174,175,180,204,205,208,210,212,215,220,221,286,299

Please improve the Results Discussion previous to conclusions

Author Response

Dear reviewer,

 Thank you for your review. Our response for your comments are as follows. If you have any new comments,please contact us.  

Response to Reviewer 2 Comments

Point 1: The manuscript must be corrected. for example In the text there are many broken links (Error! Reference source not found). In lines 80,85,92,93,104,105,106,117,130,131,141,142,149,151,154,158,163,169,171,172,174,175,180,204,205,208,210,212,215,220,221,286,299

Response 1: We are sorry for my poor mistake. And we corrected the mistake.

Point 2: Please improve the Results Discussion previous to conclusions

Response 2: authors improved the simulation discussion. Please see the revised.

Reviewer 3 Report

The paper describes application of the dynamic PLS (partial least squares) model in the MPC structure with state-space modeling. Both the dynamic PLS formulation and the MPC formulation are standard.

My main comments:

To justify the novelty of the approach the authors should explain, at least adequately comment or show in simulations, the range of applications where this type of modeling can be superior for MPC when compared with standard state-space modeling.

The authors apply state estimation (full state not measurable) within the MPC structure claiming that there are two approaches there in the case with constant disturbances: the use of state-space model in velocity form or the use of extended state-and-disturbance modeling. This is not true, as the third, competitive approach was proposed in 2014 by Tatjewski (Int. J. Applied Mathematics and Computer Science, Vol. 24(2), 313–323, DOI: 10.2478/amcs-2014-0023). The authors should be aware of that and should include also this technique into their comparison.

Author Response

Dear reviewer,

 Thank you for your review. Our response for your comments are as follows. If you have any new comments,please contact us.  

Response to Reviewer 3 Comments

Point 1: To justify the novelty of the approach the authors should explain, at least adequately comment or show in simulations, the range of applications where this type of modeling can be superior for MPC when compared with standard state-space modeling.

Response 1: In study case 1. A comparison of the model order between dynamic PLS model and conventional state space model is added. One can see that the inner model in dynamic PLS model is much simpler thant conventional state space model. The higher the system dimension is , the more obvious this advantage is.

Point 2: The authors apply state estimation (full state not measurable) within the MPC structure claiming that there are two approaches there in the case with constant disturbances: the use of state-space model in velocity form or the use of extended state-and-disturbance modeling. This is not true, as the third, competitive approach was proposed in 2014 by Tatjewski (Int. J. Applied Mathematics and Computer Science, Vol. 24(2), 313–323, DOI: 10.2478/amcs-2014-0023). The authors should be aware of that and should include also this technique into their comparison.

Response 2: Thank you for your good reference by Tatjewski. In Tatjewski’s paper, he proposed three methods for offset free control. One of them proposed in Section 3 is MPC with a state-space model and a measured state. Tatjewski pointed out that the precondition of this method is that state is measured. As I mentioned in Section 3.1 in my paper, the mismatch error is unavoidably existed. That is the main reason for state error. While, error  is unmeasured in dynamic PLS framework. So, I’m afraid that this method is not suitable for dynamic PLS framework.

Round 2

Reviewer 3 Report

The authors copletely ignored my second and main comment. They write that in the mentioned reference the case with measured state only is considered, which is completely not true. The main part of the mentioned paper  (the main Section 4) is devoted to the case with state estimation, by Luenberger observers or Kalman filter, and the approach presented there is therefore perfectly suited for the dynamic PLS framework. The authors even did not include the mentioned reference into the reference list! As the mentioned paper presents "state of the art" of the  MPC with state-space model and estimated state, it should definitely be included into the paper and the method taken into account.

Author Response

Dear Reviewer 3

   We apologize for our poor response in the first round of revised manuscript. Your comments are very correct. In Tatjewski’s paper, the method in section 3 is MPC needs measured state. And in section 4, filter is introduced and suitable for the case of unmeasured state. That’s what is need for dynamic PLS framework.

    We apologize again for our mistake that did not included Tatjewski’s paper in our paper. We corrected the mistake. Please see the new revised manuscript.

Round 3

Reviewer 3 Report

The authors included a short comment on the mentioned reference by Tatjewski (2014), but they seem they did not read this reference as the comment is not correct. The "third method" in Tatjewski's paper (as described by the authors) is not the extended velocity form  state-space model, but the new method with original state disturbance model and estimation of the process state (not extended). Also the incomplete reference description in the References seem to confirm my statement from the first sentence of this comments. I advice the authors to read that paper (Int. J. Appl. Math. Comput. Sci., 2014, Vol. 24, No. 2, 313–323, DOI: 10.2478/amcs-2014-0023) as it is vital for the offset-free control with state-space models  and, at least, to comment it properly. A better solution would be, certainly, to include this method into the simulations and comparisons.

Author Response

Dear Reviewer, thank you for your review. Our response for your comments are as follows.

(1)     We apologize for our description about the “third method” in Tatjewski's paper in the form of “extended velocity form state-space model”. In our opinion, Tatjewski defined the state-sapce equation in equation (48) and the extended state vector in equation(49), they are different from the conventional form of state-space model. So we thought it is “extended velocity form state-space model”. We have discussed your review carefully. May be this discription is not correct in English. And it would be more corrrectly written as “the third is with velocity model which is extended from conventional state-space model.”

(2)     We have corrected the format error of the reference.

(3)     Thank you for your advice to include Tatjewski's method into simulation and comparison. In our manuscript, we focus on solving the problem of offset-free control in the dynamic PLS framework. This problem is cuased by PLS modeling method. The simulations focus on comparing the proposed method and conventional method both in dynamic PLS framework. That is to prove the efficiency of solving the problem of offset-free control in the dynamic PLS framework.

Although, Tatjewski's methods focus on solving the offset-free control problem. But, the reason for this problem is different from that for dynamic PLS framework. There is a special reason for dynamic PLS, that is iterative principle. So we think that the advantages of the proposed method can’t be outstanding by comparision between Tatjewski's methods and the proposed methods.

Thanks again for your advices. And we will make more efforts to sovle more general offset-free control problem in future work, not only in dynamic PLS framework.

If you have any new comments, please contact us.